# Echovirus-30 Infection Alters Host Proteins in Lipid Rafts at the Cerebrospinal Fluid Barrier In Vitro

**DOI:** 10.3390/microorganisms8121958

**Published:** 2020-12-10

**Authors:** Marie Wiatr, Simon Staubach, Ricardo Figueiredo, Carolin Stump-Guthier, Hiroshi Ishikawa, Christian Schwerk, Horst Schroten, Franz-Georg Hanisch, Henriette Rudolph, Tobias Tenenbaum

**Affiliations:** 1Pediatric Infectious Diseases, University Children’s Hospital Mannheim, Medical Faculty Mannheim, Heidelberg University, 68167 Mannheim, Germany; carolin.stump-guthier@medma.uni-heidelberg.de (C.S.-G.); christian.schwerk@medma.uni-heidelberg.de (C.S.); horst.schroten@umm.de (H.S.); henriette.rudolph@kgu.de (H.R.); Tobias.tenenbaum@medma.uni-heidelberg.de (T.T.); 2Institute for Transfusion Medicine, University Hospital of Essen, University of Druisburg-Essen, 45157 Essen, Germany; Simon.Staubach@uk-essen.de; 3GenXPro GmbH, 60438 Frankfurt am Main, Germany; rfigueiredo@genxpro.de; 4Johann Wolfgang Goethe University Frankfurt, 60438 Frankfurt am Main, Germany; 5Laboratory of Clinical Regenerative Medicine, Department of Neurosurgery, Faculty of Medicine, University of Tsukuba, Tennodai, Tsukuba, Ibaraki 305-8575, Japan; ishi-hiro.crm@md.tsukuba.ac.jp; 6Institute of Biochemistry II, Medical Faculty, University of Cologne, 50931 Cologne, Germany; franz.hanisch@uni-koeln.de

**Keywords:** enterovirus, Echovirus-30, lipid raft, blood-cerebrospinal fluid barrier, viral infection, HIBCPP cells

## Abstract

Echovirus-30 (E-30) is a non-polio enterovirus responsible for meningitis outbreaks in children worldwide. To gain access to the central nervous system (CNS), E-30 first has to cross the blood-brain barrier (BBB) or the blood-cerebrospinal fluid barrier (BCSFB). E-30 may use lipid rafts of the host cells to interact with and to invade the BCSFB. To study enteroviral infection of the BCSFB, an established in vitro model based on human immortalized brain choroid plexus papilloma (HIBCPP) cells has been used. Here, we investigated the impact of E-30 infection on the protein content of the lipid rafts at the BCSFB in vitro. Mass spectrometry analysis following E-30 infection versus uninfected conditions revealed differential abundancy in proteins implicated in cellular adhesion, cytoskeleton remodeling, and endocytosis/vesicle budding. Further, we evaluated the blocking of endocytosis via clathrin/dynamin blocking and its consequences for E-30 induced barrier disruption. Interestingly, blocking of endocytosis had no impact on the capacity of E-30 to induce loss of barrier properties in HIBCPP cells. Altogether, these data highlight the impact of E-30 on HIBCPP cells microdomain as an important factor for host cell alteration.

## 1. Introduction

Enterovirus infection outbreaks occur every year worldwide especially in the Asian-Pacific region. Amongst the most common epidemies, Echovirus-30 (E-30) infection in children rises every year in high and especially in low-income countries, which re-enforces the economic difficulties of the southern region [1]. Enteroviruses usually reside in the digestive tract, however, they can enter the brain and cause neurological damages during a viremic phase [2,3]. Entry of pathogens into the brain can lead be lethal [4,5]. In order to create an inflammation of the meninges and the brain, E-30 first has to breach one of the blood-brain barriers such as the blood-cerebrospinal fluid barrier (BCSFB) [6,7]. The BCSFB is located at the choroid plexuses in the brain and is composed of epithelial cells tightly sealed together, providing a high transepithelial electrical resistance (TEER) and a low permeability for macromolecules [8,9]. Among various properties, adherens junction components such as cadherin and tight junction components such as ZO-1, claudins, and occludin are in active contact with the actin fibers of the epithelial cells, which is essential for the maintenance of the barrier function [10,11,12]. The epithelial cells at the choroid plexus are also rich in transporters and carrier receptors, such as transferrin receptor (TfR), the water channel aquaporin (AQP1), and sodium channels, which provide elements that are necessary for preserving the brain equilibrium [11,13,14]. The compartmentalization by junctional proteins is essential as polarization of the epithelial cells in the choroid plexus is highly pronounced compared to the BBB [15]. The only functional human in vitro model of the BCSFB is the cell layer formed by HIBCPP (Human Immortalized Choroid Plexus Papilloma) cells [16]. Due to its strong barrier properties such as high TEER and low permeability, HIBCPP cells are a solid model to study viral infection and its impact on the BCSFB [17,18].

The underlying mechanisms used by enteroviruses to enter the brain and affect the barrier function remain not fully understood [7,19]. Nevertheless, the lipid raft or microdomains of the cells are potential docking structures that many viruses can use to enter a cell [20,21]. The proteins and receptors present in the microdomains facilitate the entry of virus via formation of vesicles [22,23]. In vitro studies revealed that echovirus-1 (E-1) and coxsackievirus B (CVB) enter epithelial colorectal adenocarcinoma (Caco-2) cells through the decay accelerating factor (DAF) and coxsackie virus-associated receptor (CAR) in a dynamin-independent pathway [24]. In contrast, to enter non-polarized cells, CVB only uses CAR in a dynamin-dependent-pathway [25]. Interestingly, the HIBCPP cell model showed that blocking of clathrin-mediated endocytosis decreased *Listeria monocytogenes* invasion of the BCSFB in vitro [26]. Furthermore, a study focused on E-30 strain *Bastianni* revealed that basolateral infection of the HIBCPP cells has a disruptive effect on tight junction proteins such as ZO-1, but also on adherens junction proteins such as E-cadherin [27]. In addition, another recent study described the importance of HIBCPP cells polarity for E-30 infection and its impact on the cells at an RNA level [28].

In this study, we focused on E-30 infection and the consequences for the proteins present at the cell membrane. We performed differential proteomics by mass spectrometry on the protein content of the lipid rafts in the HIBCPP cells following E-30 infection. We found that E-30 infection led to a reduced content of proteins implicated in cellular organization. Hereafter, we performed blocking experiments by inhibiting clathrin-coated endocytosis, a mechanism that takes place in the lipid raft, and evaluated the consequences for E-30 induced barrier disruption in HIBCPP cells. Blocking of clathrin- and dynamin-mediated endocytosis had no consequences for E-30 and its ability to alter HIBCPP cell barrier function. Altogether, our data show that E-30 infection in a BCSFB in vitro model leads to protein loss at host cell microdomains.

## 2. Materials and Methods

### 2.1. Human Choroid Plexus Papilloma (HIBCPP) Cell Culture

HIBCPP cells were cultured in a T175 flask (Greiner, Germany) in HIBCPP 10% medium consisting of Dulbecco’s modified eagle medium (DMEM)/Ham’s F12 1:1 medium (Life Technologies, UK) supplemented with 4 mM L-Glutamine, 5 μg/mL insulin, 10% fetal calf serum (FCS) (Gibco, USA), and 2.5% pen/strep (MP Biomedicals, Santa Ana, CA, USA). When HIBCPP cells reached confluence, they were further seeded on filter inserts (Cell inserts Sarstedt, Germany; pore diameter 5.0 μm, pore density 6.0 × 105 pores/cm^2^, 0.33 cm^2^) in standard (CytoOne© 24 well plate StarLab, Milton Keynes, UK) culture model. HIBCPP cells that reached a TEER of [50 Ω × cm^2^] were further switched to HIBCPP 1% medium consisting of DMEM/Ham’s F12 1:1 medium, supplemented with 4 mM L-glutamine, 5 μg/mL insulin, and 1% heat inactivated FCS. Cells were ready to use for experiments 24 h after this last medium change.

### 2.2. Infection of HIBCPP Cells with E-30

Echovirus 30 (E-30) strain *Bastianni* was obtained from the National Reference Center for Poliomyelitis and Enteroviruses (NRCPE) at the Robert Koch institute RKI (Berlin, Germany). Once HIBCPP cells reached a TEER between [215–775 Ω × cm^2^], they were infected with E-30 at MOI 20. The concentration of the E-30 stock was determined using quantitative TaqMan real-time PCR analysis. In short, to infect one insert with HIBCPP cells with a MOI of 20, we considered the number of HIBCPP cells as a fixed factor of 400.000 cells/insert, and we diluted 2.48 µL of E-30 *Bastianni* stock solution in 400 µL HIBCPP 1% medium in inserts.

### 2.3. Evaluation of the Barrier Integrity

The barrier integrity of the HIBCPP cell layer was evaluated via TEER measurement with a tissue voltohmmeter (Milicell© ERS-2 Epithelial Volt-Ohm meter, Millipore, Germany). To perform experiments, HIBCPP cells had to reach a high TEER [215–775 Ω × cm^2^]. The paracellular flux of HIBCPP cells was quantified with Dextran-TexasRed© (Ex 595/Em 615) (Invitrogen, Germany) tracer solution, which has a molecular weight of 1000 Da. The tracer solution (5 μL) was added to the upper part of the well containing 400 µL of HIBCPP 1% medium. After 4 h incubation, the bottom part of the well was stirred and the percentage of Dextran-TexasRed© was further quantified via fluorescence measurement with a TECAN 200 M Infinite Multiwell reader (Tecan, Männedorf, Switzerland). A concentration curve was generated to determine the percentage of Dextran-TexasRed©, which correlates with the permeability of the cells.

### 2.4. Immunofluorescence of HIBCPP Cells

At the end of the experiment, HIBCPP cells inserts were rinsed with PBS (Gibco, Thermofisher, Waltham, MA, USA) and fixed in 3.7% formaldehyde for 15 min at room temperature (RT). Cells were further washed with PBS, membrane culture inserts were cut out and permeabilized with 1% Triton-X-100 PBS, at RT for 20 min. The HIBCPP cells were further incubated in 1% BSA/PBS solution for 15 min at RT, to block unspecific binding sites. Further, incubation with the primary antibody, monoclonal mouse light diagnostics™ anti-PAN Enterovirus (1:250) (Merck, Germany), was performed overnight at 4 °C. The following day, cells were washed with PBS and incubated with a secondary antibody, Alexa Fluor anti-mouse 568 as well as 4′-6-diamidino-2-phenylindole dihydrochloride (DAPI), diluted 1:50.000 in 1% BSA/PBS at RT for 1 h. After this incubation, cells were washed in PBS and mounted with anti-fade reagent (Life Technologies, Carlsbad, CA, USA).

### 2.5. E-30 Viral Particle Quantification in HIBCPP Cells

Following 24 h of E-30 infection MOI 20 in HIBCPP cells, 200 µL of each compartment was stirred and collected. To isolate the viral copies, we used the EZ1 Virus Mini Kit v2.0 (Qiagen, Germany), and the fully automated system EZ1 Advanced XL, following the instructor’s protocol. Calculation of the viral copies/mL was performed as follows: RNA was extracted from 200 µL samples and eluted with 60 µL of elution buffer. Finally, 2.5 µL of the eluate were used for the PCR, which gave 1/24 of complete RNA isolated. The number of copies/PCR reaction was calculated by the standard curve and multiplied by 24. To achieve the number of copies/mL, the total number of copies/PCR reaction was multiplied by 5. The number of copies/PCR reaction obtained by the q-PCR instrument was multiplied by 120. This gave the final number of viral genome copies per mL.

### 2.6. Lipid Raft Isolation of HIBCPP Cells

The lipid raft isolation was performed as previously published in [29]. In order to prepare lipid rafts, HIBCPP cells were seeded in a T175 cm^2^ flask in HIBCPP 10% medium. When they reached confluency, HIBCPP cells were harvested using trypsin (Figure 1). Cell samples from two different flasks (T 175 cm^2^) were pooled together in warm PBS (37 °C) containing trypsin inhibitor (Applichem, Germany), followed by centrifugation for 5 min at 1000 rpm. Next, cell pellets were washed twice with PBS at 4 °C. The pellets were then adjusted with cold PBS up to 1.6 mL. A volume of 200 µL of protease inhibitor (Roche Protease Inhibitor tablet solved in 1 mL PBS) was added, and HIBCPP cells were dispersed by gentle vortexing. Another volume of 200 µL Triton X-100 (10%) was added to obtain a final volume of 2 mL containing 1% Triton X-100. Subsequently, the samples were rotated for 1 h at 4 °C. Afterwards, 1.8 g of sucrose was added to the homogenate, then filled up to a volume of 4 mL (final sucrose concentration, 45%). The samples were rotated for another 30 min at 4 °C before being transferred into ultra-clear centrifuge tubes (Beckman Instruments, Fullerton, CA, USA). After this step, 3 mL of 35% sucrose solution and 1.5 mL of 5% sucrose solution were layered stepwise on top. Buoyant density centrifugation was performed at 198.000× *g* with SW41 rotor/Beckman for 2.5 h at 4 °C, in order to make the lipid rafts float. A volume of 3 mL corresponding to the light scattering band that contained the lipid rafts was collected and then centrifuged at 114.000× *g* for 1.5 h, promoting the lipid raft fraction to be pulled down (Ti-60 rotor/Beckman). Afterwards, the pellets were transferred into 1.5 mL of cold 2% SDS/PBS in Beckmann tubes and further centrifuged at 40.000 rpm with TLA55 rotor/Beckmann. Finally, the lipid rafts were added into 1.5 mL Eppendorf tubes, and chloroform-methanol-water extraction was performed (Figure 1).

### 2.7. Sample Preparation for Mass Spectrometry

Protein pellets were pooled in 20 mL urea (8 M) and 4 µL protease-inhibitor cocktail, in ammonium hydrogen carbonate (5×). Dithiotreitol (1.2 µL, 100 mM) was added and the samples were incubated for 1 h at 37 °C. Alkylation was performed by adding 1.8 µL chloroacetamide (40 mM) for 30 min at RT in the dark. After digestion with lysyl-enopeptidase (Lys-C) for 3.5 h at 37 °C, the samples were diluted in 60 µL of ammonium hydrogen carbonate (50 mM, pH 8.0) to reduce urea concentration (<2 M). Proteins were further digested with 0.5 µg of trypsin (porcine, Promega, Madison, WI, USA) at 37 °C overnight. Prior to sample work-up by solid-phase extraction onto ZipTip-C18, the peptide mixture was acidified with 10 µL of 10% formic acid.

### 2.8. Ultra-High-Performance Liquid Chromatography Electrospray Ionisation Tandem Mass Spectrometry (UHPLC-ESI-MS/MS Analysis)

Samples were run on a Q-Exactive Plus Orbitrap (Thermo Scientific, Waltham, MA, USA) equipped with EASY-nLC 1000 UHPLC. Separation of peptides was performed on an Acclaim PepMap RSLC 150 mm C18 column, 50 µm of internal diameter (2 µm bead diameter, 100 A) with a flow rate of 200 nL/min and in a gradient of 0.1% formic acid in water (buffer A) and 0.1% formic acid in acetonitril (buffer B). Content of buffer B was increased from 2% to 30% during 240 min and from 40% to 100% in 5 min. The sample load was 5 µL. The mass spectrometer was operated in the data dependent mode with automatic switching between full scan MS and MS/MS acquisition. Survey full scan MS spectra (m/z 300−1800) were acquired in the Orbitrap with a resolution of 70,000 (m/z 200) after accumulation of ions to a target value of 3 × 10^6^, based on predictive AGC from the previous full scan. Dynamic exclusion was set to 20 s. The 12 most intense multiply charged ions (z ≥ 2) were sequentially isolated and fragmented in the octopole collision cell, by higher-energy collisional dissociation (HCD) with a maximum injection time of 120 ms. A range from 50 to M + 50 Da was covered for MS2 (resolution of 17,500). A 2.5 Da isolation width was chosen.

### 2.9. Clathrin/Dynamin Blocking Experiment on HIBCPP Cells

In order to perform blocking experiments, we first diluted Dynasore©, chlorpromazine, and Pitstop-2© in DMSO (Figure 2) [19]. For the experiment, the compounds were further diluted in HIBCPP 1% medium at different concentrations based on previous publications [19]. At first, inverted cultures of HIBCPP cells were incubated with Dynasore© at 80 µM and 120 µM, and chlorpromazine at 30 µM, and Pitstop-2© at 60 µM for 30 min at 37 °C/5% CO_2_. We also used a negative control for Pitstop-2© and chlorpromazine, which consisted of empty molecules with the same structures, but deprived from active sites. For Dynasore©, DMSO was used as control. The blocking compounds were added on the HIBCPP cells at T = 0 h. After 30 min of incubation, HIBCPP cells were infected with E-30 at MOI 0.7 in HIBCPP 1% medium. Following 24 h of infection, Dextran-Red© was added to the upper compartment of the insert and TEER was measured as previously described. Additionally, 24 h post-infection, dextran was added to the basolateral side of HIBCPP cells as described in *2.3*. At the end of the 28 h of infection with E-30, TEER was measured and Dextran-Red© quantified as described in *2.3*. Live/Dead viability assays were performed using the LIVE/DEAD^®^ viability/cytotoxicity kit (Thermofisher, Bremen, Germany), on HIBCPP cells following the manufacturer’s instructions.

### 2.10. Data Processing and Statistics

Raw mass spectrometry (MS) files were processed with MaxQuant, in the Perseus framework (MaxQuant version 1.5.3.8) software suite. Peak list files were searched by the Andromeda search engine and subsequently incorporated into the Max-Quant framework. The database searching parameters included: enzyme name: Trypsin (full), maximum, missed cleavage sites: 2, minimum and peptide length: 7. Variable modifications were: methionine oxidation, and peptide N-terminus acetylation. Fixed modification was: cysteine carbamidomethylation. A threshold value of values of false discovery rate (FDR) of 0.01 was used for proteins and peptides. Processed data was quantitatively evaluated in Perseus (version: 1.5.2.4). Ratios were transformed to log2 values and normalized by subtraction of the medians. Proteins were identified as significantly changed in abundancy, when they exceeded a *t*-test difference < −1.0 or >1.0 and a *t*-test *p*-value lower than 0.01. Significantly fold-changed proteins were searched for affected biological processes with the Gene Ontology enrichment tool (GOTE), which is provided by the Gene Ontology Consortium.

STRING interaction analysis was performed with Cytoscape software version 3.8.0, and STRINGApp, considering a high confidence score for interactions [30].

For all the other experiments, statistical analysis was conducted using the Student’s *t*-test with GraphPad Quickcalcs online software (GraphPad Software, San Diego, CA, USA). Each one of the conditions, uninfected or infected with E-30 MOI 20, treated or untreated with lipid raft inhibitor, were considered as a fixed factor. Each mean per condition was compared two by two with the Student’s *t*-test. The figures are represented as mean ± SD.

### 2.11. Availability of Data

The raw proteomic data files were deposited into ProteomeXchange repository PXD001890.

## 3. Results

### 3.1. Infection of HIBCPP Cells with Echovirus-30 MOI 20

Our previous results suggested that the optimal infection rates of HIBCPP cells with E-30 were obtained when the cells were challenged apically with a MOI (multiplicity of infection) of 20 [28]. Now, we further analyzed several parameters, such as the transepithelial resistance, the cytotoxic effect of E-30, the number of viral particles per HIBCPP cells, and the percentage of infected HIBCPP cells (Appendix A). We observed a drastic decrease in the TEER of HIBCPP cells infected with E-30 at MOI 20 for 24 h compared to the uninfected control (** *p* < 0.001) (Appendix A). Next, we quantified E-30 viral particles contained in each compartment of the culture system, on filter insert at T = 0 h and at T = 24 h of infection with E-30 as well as of the uninfected control. The number of viral copies present in the HIBCPP cells at T = 24 h was high with more than 5 × 10^6^ copies per cell. Furthermore, the number of viral copies present in the top and bottom compartments at the end of the 24 h of infection was also elevated with more than 1.5 × 10^7^ copies in the top compartment and 2.5 × 10^7^ viral particles present in the bottom compartment of the culture system (Appendix A). We further assessed possible cytotoxic effects of E-30 infection on the viability of HIBCPP cells via live/dead assay. The number of dead cells remained low 24 h post-infection with E-30 MOI 20 when compared to uninfected conditions (Appendix A). As shown in the immunofluorescence pictures, there are no clear differences in the infected versus uninfected HIBCPP layer. Lastly, we evaluated the number of HIBCPP cells infected at 24 h with E-30 by immunofluorescence. We observed a high percentage of infected HIBCPP cells 24 h post-infection at MOI 20 (Appendix A). Altogether, these data show that E-30 infection with an MOI 20 for 24 h is well suited to perform a proteomic assay on the protein composition of the lipid rafts of HIBCPP cells, since these infection conditions have a high impact on the HIBCPP cell properties, lead to a high number of infected cells and a high amount of viral particles, as well as a low cytotoxicity in HIBCPP cells.

### 3.2. Differential Abundancies of Proteins in the Lipid Raft Composition of HIBCPP Cells Following E-30 Infection

In the next step, we infected HIBCPP cells with E-30 at MOI 20 for 24 h and performed the lipid raft isolation, and further analyzed the proteins present in the lipid rafts by mass spectroscopy. At first, we confirmed the presence of typical lipid rafts proteins in our samples (Figure 3). A high proportion of proteins such as FLOT1 flotillin-1, integrins such as ITG4, ITGα3, ITGβ1, ITGα2, and heat shock proteins such as HSPA1B and HSP90B1 were present in our samples (Figure 3).

Then, we analyzed the proteins present in the lipid rafts by mass spectroscopy. Hierarchical clustering and principal component analysis (PCA) were used to assess relatedness between samples and grouped the expression profiles from infected and uninfected control conditions in two separate clusters (Figure 4a). From this analysis, we observed 224 proteins with differential abundancies, including 82 down-regulated proteins and 142 up-regulated proteins in the lipid rafts, comparing uninfected versus E-30 infected HIBCPP cells (Figure 4c). Following E-30 infection of HIBCPP cells, we observed an up-regulation of several interferon-induced proteins such as IFI3 and IFIT1 (Figure 4c, Appendix A), implicated in innate immune system reaction and cellular communication. Interestingly, these pathways were significantly up-regulated in HIBCPP cells after E-30 infection compared to uninfected control (Figure 5a). In addition, MVP was down-regulated and OAS2 was up-regulated in the lipid rafts of infected HIBCPP cells, both implicated in innate immune response following viral infection (Figure 5b). We also noticed an up-regulation of CEACAM-1 and RHOC, which are implicated in cell–cell adhesion and desmosome organization (Figure 4b). The cell–cell adhesion and desmosome organization pathways were significantly down-regulated in HIBCPP cells following E-30 infection compared to uninfected control (Figure 5b).

Interestingly, the tight junction proteins ZO-1 and ZO-3 were down-regulated in the lipid rafts of infected HIBCPP cells (Appendix A). These proteins are involved in cell junction assembly, a pathway that was significantly down-regulated in E-30 infected HIBCPP cells compared to uninfected (Figure 5b). Additionally, RAP1 and ALG8, which are part of the cellular organization pathway were down-regulated in HIBCPP cells following E-30 infection (Figure 4b and Figure 5b). Furthermore, proteins such as PARP4, DNAJA1 were down-regulated and DHCR24 was up-regulated (Figure 4b). These proteins are playing a role in apoptosis and mitochondrial activity, two pathways impacted by E-30 infection. One of the pathways in which these proteins are implicated, inner mitochondrial membrane organization, was significantly up-regulated following E-30 infection. Clathrin-mediated endocytosis was significantly up-regulated in E-30 infected HIBCPP cells, due to the up-regulation of ANKYF1, ACBD3, M6PR, RAB10 and the down-regulation of TPD52 and CLTA proteins in E-30 versus uninfected cells (Figure 4b). Interestingly, biosynthetic pathways and glucose metabolic process were significantly up-regulated in the infected cells compared to the uninfected condition (Figure 5a). The cytoskeleton organization pathway was significantly down-regulated, consequently to the down-regulation of RADIXIN, PFDN6, DYNLB1, and BRK1 proteins in E-30 infected HIBCPP cells compared to the control (Figure 4b and Figure 5b).

Taken together, these results show that E-30 infection of HIBCPP cells promotes an alteration of the lipid raft organization via down-regulation of proteins responsible for cell–cell adhesion and cytoskeleton organization. However, infection with E-30 induced an increase of proteins involved in cell communication, metabolic processes, and mitochondrial membrane reorganization.

A previous study of our laboratory on E-30 infection of the HIBCPP with MOI 20 revealed a strong impact of the virus at the RNA level [28]. We compared the differentially expressed genes after E-30 infection in HIBCPPP with the new proteomic results. Among the 308 differentially expressed genes and the 224 differentially abundant proteins in the lipid raft, we could identify three genes/proteins up-regulated following E-30 infection in both assays; IFIT1, IFIT3, and OAS2 (Figure 6).

To explore further the interaction of these proteins, we performed a STRING analysis with the Cytoscape software to show the mutual relations among identified proteins (Figure 7). We observed that IFIT1, IFIT3, and OAS2 have strong interactions. In addition, CLTB CLTA, AP1B1, and M6PR are interacting with each other. Many solute carrier proteins such as SLC25A4, SLC25A5, and SLC25A3 were interacting. Further, we saw that CEACAM was interacting with PLAUR and DYNLL1, and RHOC was associated with ACTR2. Several proteins related to the actin cytoskeleton, ACTR and ARPC were forming a big cluster of interaction in the STRING network. The same phenomenon was observed for proteins implicated in mitochondrial activity such as the MRP and MRPL proteins and the solute carrier proteins SLC. Interestingly, TJP1 or ZO-1 interacted strongly with CTTN (Figure 7).

### 3.3. Blocking of Clathrin-Mediated Endocytosis Does Not Impede E-30 Mediated Loss of the HIBCPP Barrier Function

As observed in the mass spectroscopy analysis, many proteins related to clathrin-mediated endocytosis were differentially abundant in E-30 infected HIBCPP cells versus the uninfected control. Therefore, to investigate the effect of endocytosis blocking on HIBCPP barrier function, we used the inverted model system to infect HIBCPP cells from the basolateral side [8,9]. As previously published, to induce the same percentage of infected HIBCPP cells with a MOI 20 on the apical side, we used a MOI 0.7 to infect the basolateral side of the HIBCPP cells [28]. We evaluated the effect on barrier disruption with different blocking drugs, such as Pitstop-2, chlorpromazine, and Dynasore (Figure 2). We monitored TEER, cytotoxicity, and permeability of the HIBCPP cells at the end of 28 h infection with E-30 and blocking. At first, treatment with Pitstop-2 ± E-30 infection had no detectable impact on HIBCPP cell viability (Figure 8a). In addition, treatment with a negative control (mock pitstop (−)) ± E-30 infection had no impact on the viability of HIBCPP cells. Only a small number of dead cells were observed, but we could not detect major differences between treated and untreated or infected versus uninfected HIBCPP layers. In contrast, a significant decrease in TEER was measured following 28 h of E-30 infection, independently of the presence or absence of Pitstop-2 or the negative control, compared to the equivalent uninfected condition (*** *p* < 0.001) (Figure 8b). Of note, the presence of Pitstop-2 had no impact on the TEER of HIBCPP cells compared to control conditions (Figure 8b). Additionally, the dextran flux significantly increased following E-30 infection regardless of the inhibition with Pitstop-2 or the negative control, when compared to the uninfected condition (*** *p* < 0.001) (Figure 8c). In the uninfected conditions, the dextran flux remained lower than 4% ± 4% whereas, when the cells were infected with E-30 in presence of inhibitor, the paracellular dextran flux reached almost 13% ± 3% (*** *p* < 0.001) (Figure 8c).

Next, we included another widely used clathrin inhibitor, chlorpromazine, to confirm our previously observed results. We saw no differences in the viability of HIBCPP cells between the negative control and the condition treated with this inhibitor in presence or absence of E-30 infection (Figure 9a). The number of dead cells was very low for all the conditions, the infection with E-30 and the presence of chlorpromazine did not impact the viability of the cells, which resulted in a very low number of dead cells for all conditions (Figure 9a). Furthermore, E-30 infection led to a decrease in TEER for all the infected conditions regardless of the presence of chlorpromazine, or for the negative control (* *p* < 0.01) (Figure 9b). The use of chlorpromazine had no impact on the TEER of the HIBCPP cells compared to control condition (Figure 9b). Lastly, the paracellular dextran flux was impacted by E-30, and we observed an increased flux upon E-30 infection (* *p* < 0.01) (Figure 8c). The latter was independent of blocking with chlorpromazine (* *p* < 0.01) (Figure 9c).

At last, dynamin blocking with Dynasore led to results similar to those obtained with Pitstop-2 and chlorpromazine (Figure 9a). Inhibition with two different concentrations of Dynasore did not influence the viability of the HIBCPP cells in presence or absence of E-30 infection (Figure 10a). Additionally, the TEER was drastically decreased following E-30 infection regardless of the presence of Dynasore (*** *p* < 0.001) (Figure 10b). The inhibition showed the same results for both concentrations (Figure 9b). Finally, the paracellular permeability was significantly increased after E-30 infection despite the presence of Dynasore (** *p* < 0.05), even with the higher concentration (Figure 10c). Taken together, these data show that E-30 could infect HIBCPP cells and, thereby, induce a decline in barrier integrity even in presence of clathrin and dynamin blocking. Neither clathrin nor dynamin blocking had an impact on cell viability or barrier function of HIBCPP cells during E-30 infection.

## 4. Discussion

In children and immunocompromised people, enterovirus infection can have a severe clinical course [2,3]. Amongst enterovirus, E-30 is one of the most common types isolated from the cerebrospinal fluid of patients with meningitis. In this study, we revealed the impact of E-30 infection in an in vitro BCSFB model at the membrane microdomain level with the down-regulation of proteins implicated in cytoskeleton remodeling and up-regulation of innate immune response pathways.

Several studies have revealed the blood-cerebrospinal fluid barrier to be an important entry site for enteroviruses into the CNS [6,7]. Membrane microdomains, and lipid rafts in particular, are primordial for membrane fluidity [22], and allow several pathogens, such as enterovirus, ebola, and Marburg virus to enter the cells [32,33]. In addition, endocytosis and formation of caveolae also take place at the lipid rafts [32]. To penetrate the BCSFB cells, E-30 may use the lipid rafts like other enteroviruses in other host cells [33]. Previous publications have shown that echoviruses take advantages of the lipid rafts structure to enter the cells [34]. For example, the lipid rafts were described as a privileged site for the CVA9 infection cycle [35], whereas E-11, E-25, and E-30 require a process dependent of the lipid raft to enter inside green monkey kidney (GMK) cell lines [33].

In this study, we used an in vitro model of the BCSFB the HIBCPP cells to study the impact of E-30 infection with a special focus on the proteins present in the lipid raft fraction of the cellular membrane. Further, we determined the consequences of inhibiting clathrin-mediated endocytosis for the impact on viability and barrier function of HIBCPP cells in context of viral infection. An MOI of 20 for 24 h was chosen for infection with E-30 on the basis of previous published studies and was well suited for a high infection without inducing cell death to perform the proteomic assay on infected HIBCPP cells [28]. The infection led to a decrease in barrier properties, a high number of infected cells, and a low cytotoxicity rate. A significant decrease in CNS barrier properties, either BBB or BCSFB, is one of the potential characteristics of changes induced by neurotropic viruses [36]. Additionally, we wanted to avoid an increase of apoptotic changes in the cells, in order to be able to define the impact of E-30 infection on cellular characteristics before appearance of the cytopathic effect induced by the virus at later time points, which has also been described previously [37].

The analysis of the lipid raft microdomains revealed an increased presence of interferon stimulated genes (ISGs), such as IFIT1 and IFIT3, which play critical roles in antiviral activity, by activating the immune defense mechanisms to further restrict the spreading of the virus in the CNS [38,39]. IFIT1 have a critical role in anti-viral activity in the CNS, e.g., by restricting viral infection of the neurotropic herpes-virus 1 [40]. Moreover, OAS2, which is involved in innate immune processes, was also up-regulated in E-30 infected HIBCPP cells compared to control. OAS2, a phospholipase protein, was shown to inactivate dengue virus by disrupting the viral envelope [41], but also restricted the generation of new viral particles in cells infected with adenovirus [42]. Interestingly, the STRING analysis revealed the close interaction between IFIT1, IFIT3, and OAS2. Furthermore, we also observed an up-regulation at the RNA level of IFIT1, IFIT3, and OAS2 in E-30 infected HIBCPP cells. Our data therefore suggest that E-30 infection leads to an up-regulation in the lipid raft of proteins implicated in innate immune defense, which are also up-regulated at a gene level.

In this work, the regulated proteins described refer only to the ones in the lipid rafts in contrast to the RNA seq analysis, which involved genes from the whole HIBCPP cells. Furthermore, if the content of the lipid rafts was analyzed at later time point, we may have seen more proteins and genes in common. In general, gene regulation does occur much quicker than protein regulation. Therefore, the genes found to be differentially regulated may not translate directly into protein alterations. This may be the reason why we have observed only a few similarities in gene and protein regulation in the comparative analysis.

CEACAM-1 protein was also up-regulated in the lipid rafts of HIBCPP cells after E-30 infection. Interestingly, the pathway in which CEACAM-1 protein plays a major role, i.e., cell–cell adhesion and cell adhesion to the extracellular matrix [43], was down-regulated. E-30 infection leads to a down-regulation of proteins such as RADIXIN, PFDN6, DYNLB1, and BRK1. These proteins were all implicated in cytoskeleton remodeling and are therefore involved in cell cohesion processes. Nevertheless, the up-regulation of CEACAM-1 was probably not sufficient to compensate the down-regulation of the latter proteins. Furthermore, other cytoskeleton related proteins which were impacted by E-30 infection such as ACTR and ARPC were forming a strong interaction network. In two studies, it was shown that virus infection had an impact on the cytoskeleton reorganization of the host cells, an event that contributes to the loss of adhesion and cell–cell interaction [44,45]. In these studies, the authors demonstrated that cytoskeleton reorganization was one of the major events for virus invasion and dissemination within the epithelial cells [44,45]. Moreover, it was previously shown that IFN-γ amplification enhanced the up-regulation of CEACAM-1 in the context of *N. meningitidis* invasion of epithelial cells [46]. CEACAM-1 may also play a role in the development of anti-viral strategies by the host. In fact, pathogenic avian influenza virus (H5N1) infection of the lung epithelium stimulates the expression of CEACAM-1, which further induced production of pro-inflammatory chemokines [47].

In our model, a loss of permeability and TEER following E-30 infection was observed, which correlates with the diminution of barrier relevant membrane proteins in HIBCPP cells such as ZO-1 and Occludin, which connect the actin cytoskeleton to the cellular membrane [27]. In our proteomic analysis, we identified a down-regulation of tight junction protein ZO-1 and ZO-3. In mice, the West Nile virus was known to induce degradation of junctional complex proteins and the subsequent disruption of the BBB [48].

The loss of epithelial cell polarity has been involved in impaired function of the BCSFB, and hence correlates with neurological disorders [49]. RAP1 and ALG8, two proteins that play a role in the maintenance of epithelial cell polarity as well as in the regulation of solute passage across epithelial cells [12,50], were also found to be differentially abundant in our proteomic assay. Previous data highlighted that polar infection of E-30 affects various genes in HIBCPP cells and involves different pathways [28]. However, due to the limitations of our protocol, we could not investigate the impact of E-30 basolateral infection on the microdomains of HIBCPP cells. Such additional proteomic assays would be of great interest in order to compare both infections and reveal their unique impacts on the protein content at the membrane of HIBCPP cells.

Many proteins involved in mitochondrial energy production, oxydo/reduction, and apoptosis such as PARP4, DNAJA1, DHCR24, and BAX were significatively impacted by E-30 infection. Furthermore, mitochondrial proteins (MRP) were forming a large network of interaction. A proteomic analysis of EV71 infected cells from a fatal case of meningitis also revealed the expression of many proteins implicated in mitochondrial-related metabolism, cell death, and cytoskeleton organization [51].

Additionally, we observed a strong interaction between ACBD, TPD52, CLTA, M6PR, and RAB10 which all play a role in clathrin-mediated endocytosis pathway. Interestingly, down-regulation of RAB GTPase in the context of enterovirus infection was observed in vivo at the BBB [52]. In addition, CLTA is mandatory for the entry of enterovirus 71 (EV71) via clathrin-mediated endocytosis, and the blocking with chlorpromazine inhibited the infection of RD cells by 80% [53]. In E-1 infection, clathrin-mediated endocytosis is also the most common pathway used by the virus to enter epithelial cells [24].

Lastly, we investigated the importance of clathrin/dynamin for E-30 induced barrier affection of HIBCPP cells. Blocking experiments demonstrated that E-30 did not require clathrin nor dynamin endocytosis pathway to infect HIBCPP cells and induce barrier affection. Interestingly, actin depolymerization was shown to disrupt the tight junction in endothelial cells via endocytosis [54]. In one study, clathrin-mediated endocytosis was required for E-7 to enter polarized Caco-2 cells, and therefore clathrin and dynamin inhibition prevented the entry of the virion in the cells [55]. Another paper revealed that increased brain barrier permeability was induced by *Clostridium perfringens* toxin via caveolae dependent transcytosis [56]. Still, our results showed that clathrin/dynamin blocking had no influence on the damage caused by E-30 on HIBCPP cells. Therefore, we conclude that neither clathrin nor dynamin seem to be important for E-30 barrier disruption of HIBCPP cells. Nevertheless, our data suggest that E-30 provokes a high dysregulation of the protein content of the lipid raft, which impact the further properties of the HIBCPP cells.

## 5. Conclusions

Infection with E-30 caused barrier affection and modification of protein in HIBCPP cells. E-30 infection of HIBCPP cell resulted in alteration of the lipid raft composition by the differential expression of proteins responsible for cell–cell adhesion, cytoskeleton organization, but also of proteins involved in cell communication, metabolic processes, and mitochondrial membrane reorganization and endocytosis/vesicle budding. However, blocking of endocytosis had no impact on the capacity of E-30 to induce loss of barrier properties in HIBCPP cells. Altogether, these data highlight the impact of E-30 on HIBCPP cells microdomain protein content. Further investigations are required to elucidate the role of specific host cell proteins in lipid rafts for viral cell entry and barrier affection.

## Figures and Tables

**Figure 1 microorganisms-08-01958-f001:**
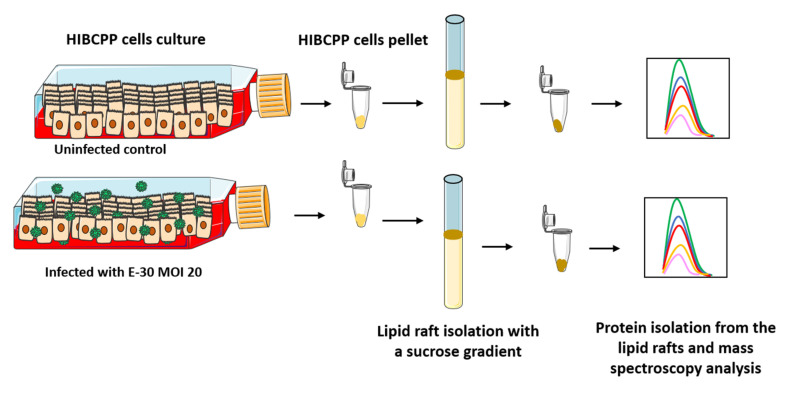
Schematic representation of the lipid raft isolation of HIBCPP (Human Immortalized Choroid Plexus Papilloma) cells. HIBCPP cells were infected with E-30 (Echovirus-30) for 24 h at a MOI (Multiplicity of infection) of 20. Subsequently, the cells were pelleted, and the lipid rafts were isolated using a sucrose gradient [29]. Finally, the protein content of the lipid raft fraction was extracted and analyzed by mass spectrometry. The HIBCPP cells are infected with E-30 on their apical side.

**Figure 2 microorganisms-08-01958-f002:**
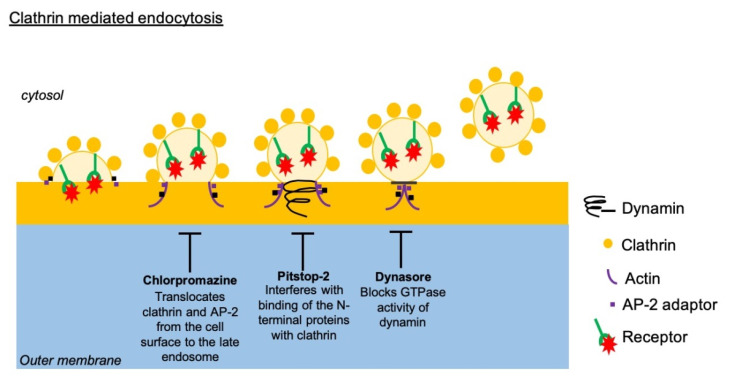
Blocking compounds affecting the clathrin-mediated endocytosis pathway. Schematic representation of clathrin-mediated endocytosis and the impact of blocking by chlorpromazine, Pitstop-2, and Dynasore. Vesicle formation is affected by chlorpromazine, Pitstop-2, and Dynasore at different levels during the formation of the clathrin-coated vesicles.

**Figure 3 microorganisms-08-01958-f003:**
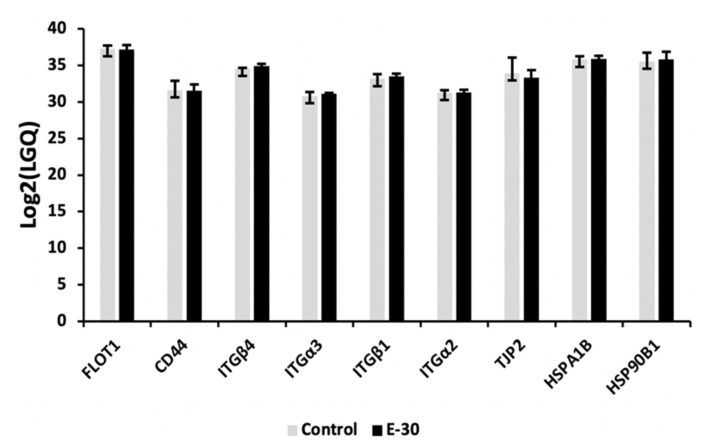
Expression of typical proteins present in the lipid raft microdomains. The graph represents label free quantification (LFQ) from lipid raft isolation of HIBCPP cells proteins in uninfected control versus E-30 infected MOI 20. The graph shows the Log2 (LFQ) mean calculated from 3 independent experiments.

**Figure 4 microorganisms-08-01958-f004:**
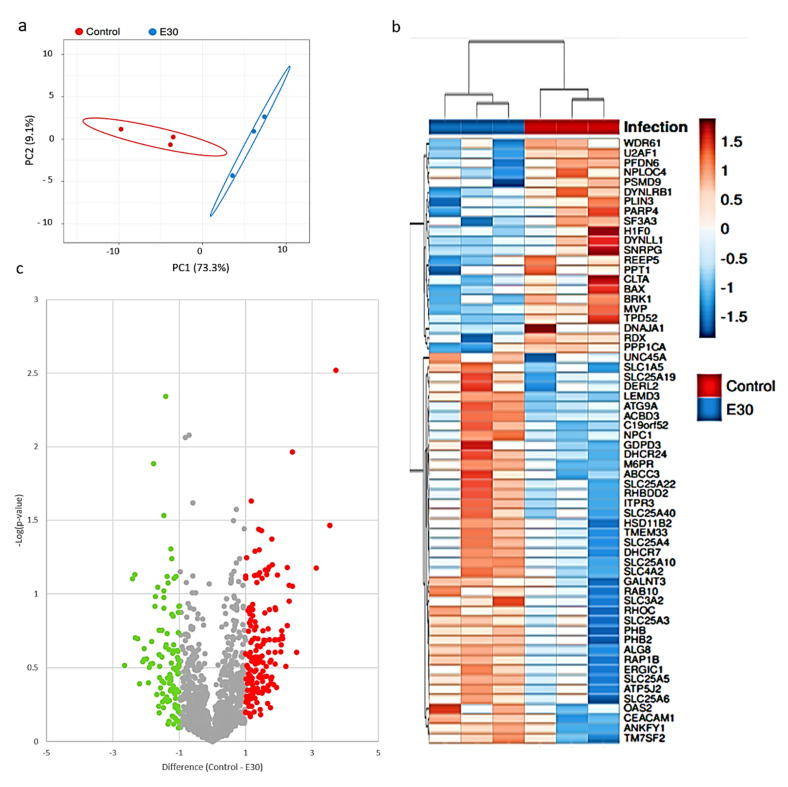
E-30 infection of HIBCPP cells alters the abundance of protein at the lipid raft micro-domains. (**a**) Plot of the principal component analysis (PCA) for the uninfected samples (red) versus E-30 MOI 20 (blue). (**b**) Heatmap representing the significantly up-regulated (in red) and down-regulated (in blue) proteins of HIBCPP cells infected with E-30 MOI 20 versus uninfected control. Proteins with a *p*-value < 0.01, and a |log2FC| > 1 were considered as differentially expressed. (**c**) Volcano plot representing proteins found to be down-regulated (in green) and up-regulated (in red) when comparing the uninfected control versus E-30 MOI 20 infected cells. Proteins in gray are not significantly changed. Proteins with a *t*-test *p*-value lower than 0.01, and an absolute difference greater than 1 were considered as differentially expressed. Statistical analysis was performed using Perseus framework (version: 1.5.2.4).

**Figure 5 microorganisms-08-01958-f005:**
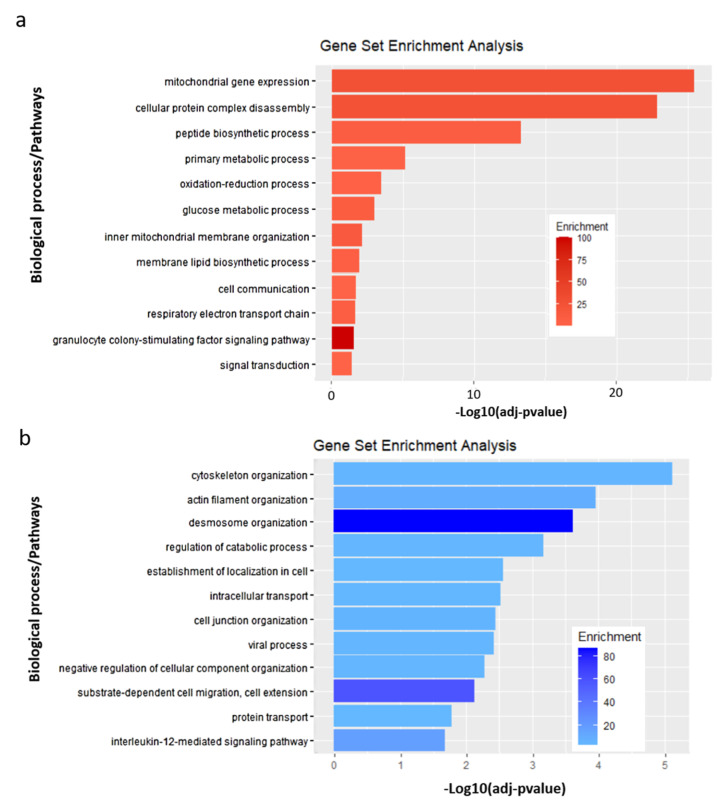
Enriched pathways in HIBCPP cells following E-30 infection versus uninfected control. (**a**) Top 12 significantly up-regulated pathways in HIBCPP cells infected with E-30 compared to uninfected control. Analysis was performed using Gene Ontology enrichment tool (GOTE). The shade of red is representative of the grade of pathway enrichment. (**b**) Top 12 significantly down-regulated pathways in E-30 infected HIBCPP cells compared to uninfected control. The shade of blue is representative of the grade of enrichment. Data are shown for 3 independent experiments. Analysis was performed using Gene Ontology enrichment tool and resulted in a binomial statistic *p*-value and pathway fold enrichment (calculated as suggested by [31]).

**Figure 6 microorganisms-08-01958-f006:**
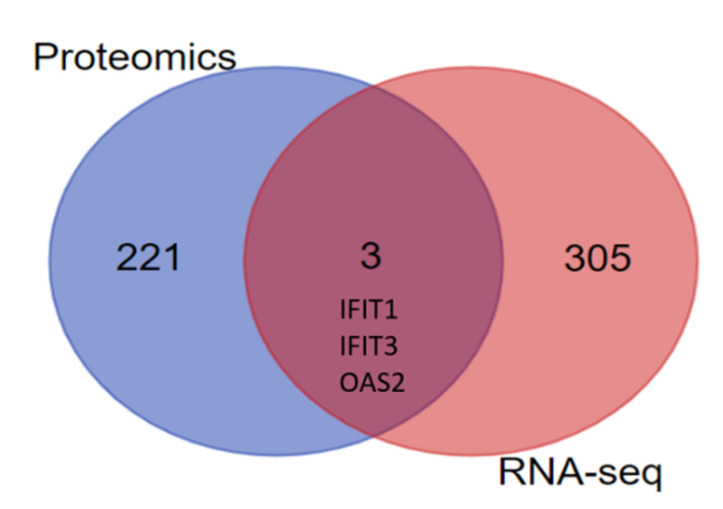
Differentially expressed genes and proteins following E-30 infection of HIBCPP cells. Venn diagram representing differentially expressed genes (308) or differentially abundant proteins (224) following 24 h of infection with E-30 at MOI 20. Among the differentially expressed genes/proteins, IFIT1, IFIT3, and OAS2 were up-regulated in both databases. Genes/proteins with *p* < 0.05, and a |log2FC| > 1 were considered as differentially expressed (*n* = 3).

**Figure 7 microorganisms-08-01958-f007:**
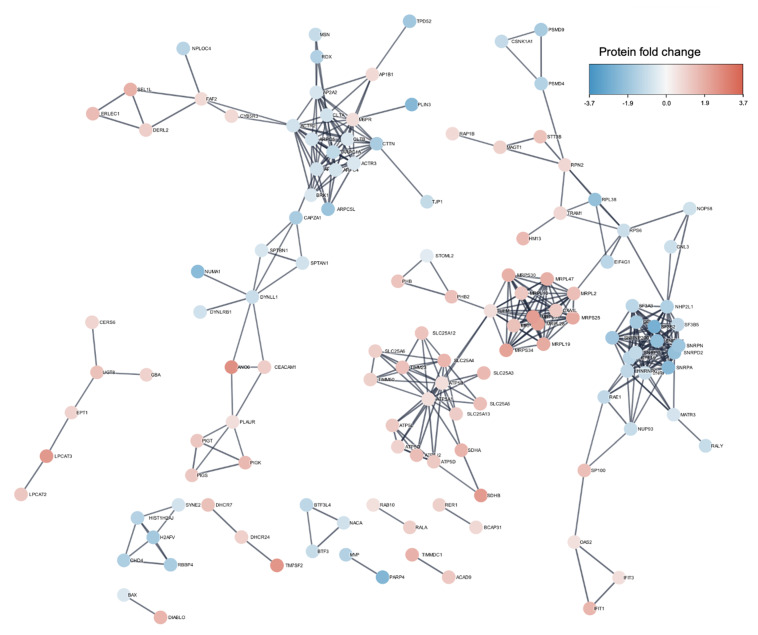
STRING interaction analysis of the 224 differentially abundant proteins using Cytoscape software. The identifiers for the 224 proteins from Appendix A were entered into the Cytoscape database showing the interaction network. The confidence level was set to high. The color indicates the protein fold change.

**Figure 8 microorganisms-08-01958-f008:**
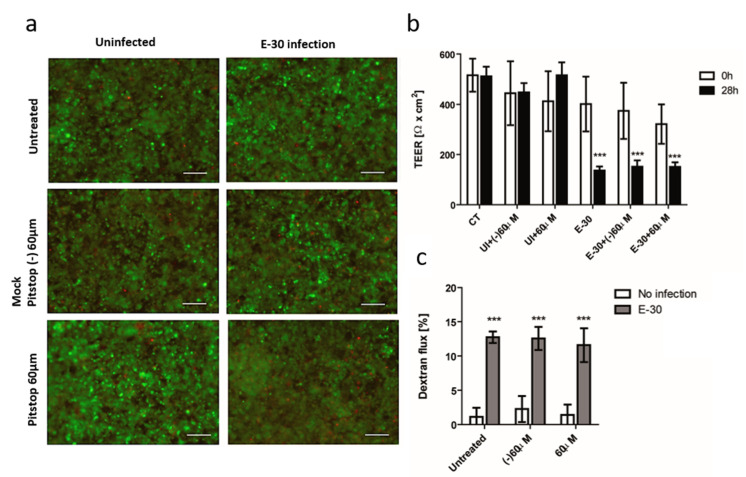
Blocking of dynamin-mediated endocytosis with Pitstop-2 has no impact on the loss of barrier properties of HIBCPP cells following E-30 infection. (**a**) Pitstop-2 and E-30 infection have no drastic impact on HIBCPP cells viability. Live/dead assay stains living cells in green and dead cells in red. Pictures show representative images from 3 independent experiments, each performed in duplicates, for the following conditions (uninfected untreated (CT), uninfected negative control of Pitstop-2 at 60 µM condition ((−)60 µM), uninfected Pitstop-2 at 60 µM condition (60 µM), E-30 infection and no blocking (E-30), negative control of Pitstop-2 at 60 µM + E-30 infection (E-30 + (−) 60 µM), Pitstop-2 at 60 µM in + E-30 infection (E-30 + 60 µM). (**b**) TEER decrease after E-30 infection in presence or absence of Pitstop-2. TEER was measured at T = 0 h (white bars) and T = 28 h (black bars) of the blocking with Pitstop-2, for all the conditions. (**c**) Increased dextran flux following E-30 infection in presence or absence of Pitstop-2. Quantification via fluorescent measurement of paracellular flux with dextran-TexasRed (MW 1000 Da), from T = 24 h to T = 28 h of blocking with Pitstop-2, for all the conditions. Statistical significance was calculated using Student’s *t*-test. *p* values are displayed as follows: *** *p* < 0.001. White scale bars represent 100 µM.

**Figure 9 microorganisms-08-01958-f009:**
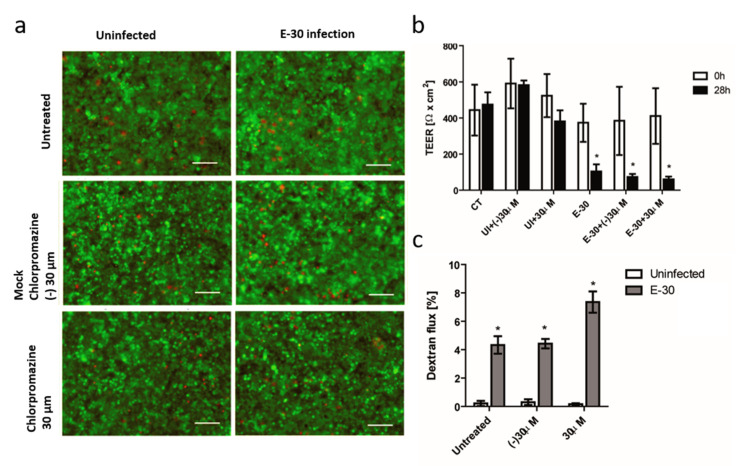
Blocking of clathrin-mediated endocytosis with chlorpromazine has no impact on the loss of barrier properties of HIBCPP cells following E-30 infection. (**a**) Chlorpromazine and E-30 infection have no drastic impact on HIBCPP cells viability. Live/dead assay stains living cells in green and dead cells in red. Pictures show representative images from 3 independent experiments, each performed in duplicates, for the following conditions; control no E-30 infection and no blocking (CT), negative control of chlorpromazine at 30 µM in uninfected condition (UI + (−) 30 µM), chlorpromazine at 30 µM in uninfected condition (UI + 30 µM), E-30 infection (E-30), negative control of chlorpromazine at 30 µM in E-30 infection (E-30 + (−) 30 µM), chlorpromazine at 30 µM in E-30 infection (E-30 + 30 µM). (**b**) TEER decrease after E-30 infection in presence or absence of chlorpromazine. TEER was monitored at T = 0 h (white bars) and T = 28 h (black bars) of the blocking with chlorpromazine, for all the conditions. (**c**) Increased dextran flux following E-30 infection in presence or absence of chlorpromazine. Quantification via fluorescent measurement of paracellular flux with dextran-TexasRed (MW 1000 Da), from T = 24 h to T = 28 h of blocking with chlorpromazine for all the conditions. Statistical significance was calculated using Student’s *t*-test. *p* values are displayed as follows: * *p* < 0.01. White scale bars represent 100 µM.

**Figure 10 microorganisms-08-01958-f010:**
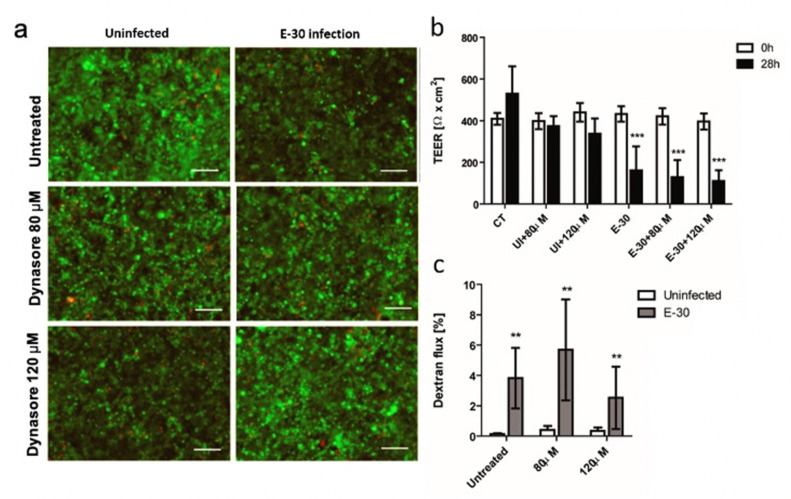
Blocking of dynamin-mediated endocytosis with Dynasore has no impact on the loss of barrier properties of HIBCPP cells following E-30 infection. (**a**) Dynasore and E-30 infection have no drastic impact on HIBCPP cells viability. Live/dead assay stains living cells in green and dead cells in red. Pictures show representative images from 3 independent experiments, each performed in duplicates for the following conditions: uninfected and untreated (CT), 80 µM of Dynasore in uninfected condition (UI + 80 µM), 120 µM of Dynasore in uninfected condition (UI + 120 µM), E-30 infection no blocking (E-30), 80 µM of Dynasore and E-30 infection (E-30 + 80 µM), and 120 µM of Dynasore and E-30 infection (E-30 + 120 µM). (**b**) TEER decrease after E-30 infection in presence or absence of Dynasore. TEER was monitored at T = 0 h (white bars) and T = 28 h (black bars) of blocking with Dynasore, for all the conditions. (**c**) Increased dextran flux following E-30 infection in presence or absence of Dynasore. Quantification via fluorescent measurement of paracellular flux with dextran-TexasRed (MW 1000 Da), from T = 24 h to T = 28 h of blocking with Dynasore for all the conditions. Statistical significance was calculated using a Student’s *t*-test. *p* values are displayed as follows: ** *p* < 0.05 and *** *p* < 0.001 was reached comparing the mean of uninfected values ± blocking with the corresponding values in E-30 infection. White scale bars represent 100 µM.

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
