# Peer review of "Echovirus-30 Infection Alters Host Proteins in Lipid Rafts at the Cerebrospinal Fluid Barrier In Vitro"

_microorganisms, 2020, doi:10.3390/microorganisms8121958_

Round 1

Reviewer 1 Report

The present study might be considered as a next step in a quest of Echovirus-30 infection accomplished by the Dr. Wiatr et al. The study experimentally done on a high level and well structured. Unfortunately, its biological impact is not high. The way the results are discussed, it represents rather a description of roles of identified proteins based on the literature data. The blocking of clathrin/dynamin pathways did not provide any impact on E-30 infection, thus, the question why proteins related to them were differently regulated upon infection remains unclear. A deeper analysis of proteomic data is required. Authors may try to perform network analysis (STRING) or use RaftProt database (https://raftprot.org/) to find the most interesting proteins, and try to understand its/their role (for example knockdown experiments). Also, I did not find any optimization steps in the section 3.1. Moreover, the outputs of this section duplicates results published in your previous paper (doi:10.3390/ijms21176268). For my mind, this part can be moved to the supplementary data as a conformation of a successful infection.

Minor points:

  1. Proteomic raw data should be uploaded to some public databases.
  2. It would be beneficial to provide interaction networks for proteins which expression was significantly changed. For example, it might help to explain why CEACAM-1 and RHOC proteins were up-regulated whereas biological processes they involved were down-regulated.
  3. Supplementary tables should be improved. Please, provide score, unique peptide number and sequence coverage. „Student’s T-test“ in the tables - does it mean significant fold change in the Log2(x) scale?
  4. More extensive comparison of proteomic data obtained in this study with gene expression profiles published earlier by the authors (doi:10.3390/ijms21176268) would be beneficial.
  5. Lines 278-281. Correct the sentence.

Reviewer 2 Report

The consequence of list of proteins changed of their expression levels after E-30 infection should be correctly revised.

1. At page 7 and 8.

MVP not abundant but decreased according to the supplemental table 2.

In addition, data regarding ZO-1 and ZO-2 could not be found by myself at supplemental Table 2. 

Please confirm and revise appropriately.

2. Regarding blocking assay of clathrin-mediated endocytosis as assessed by using inhibitors, as Chlorpromazine , Pitstop-2 and Dynsore. The manuscript used only one concentration of those drugs. Were they exactly appropriately operative. Not only dependent on previous studies, the authors should present the positive inhibitory actions of those drugs, such as caring the dose or another inhibition operating system.

Round 2

Reviewer 1 Report

The manuscript sounds better after made corrections. But the part concerning a comparison of proteomic and transcriptomic/genomic data requires more clarification. Why in the whole cell were altered only 308 genes upon infection, whereas 224 proteins were found differently regulated just in the lipid rafts. Why is the interception between genomic and proteomic data so small (only 3 entries)? Does it mean, that expression of genes coding identified proteins was not changed?

The STRING analysis must be improved. Provided network (fig.7) is complicated for understanding. It is better to use a clustering function to simplify it. Also, it would be beneficial to add fold changes of proteins, thus, it will be easier to understand the network visually. I would recommend performing STRING analysis using Cytoscape software (free of charge), and a tutorial might be found on YouTube.

Lines: 278-281. The sentence is confusing. Why do you have the same number of up- and down-regulated proteins in infected and uninfected samples?

Lines: 345-346: “. Many solute carrier proteins (SLC) were found to be strongly interacting”. This sentence should be clarified, because it is possible to describe any protein with the same words.

Please, proofread the made changes, there are some mistakes:

Line: 296: “HCR24 was up-regulating” should be up-regulated

Line 339: “proteins differentially abundant proteins”

Line 344: “strong interaction” should be interactions

Line 477: “has a been”

Line 494: “Few publications already showed”. Rephrase in passive voice

Line 514: “different pathways and [28]”

Reviewer 2 Report

The manuscript has been well revised and I feel this version is sufficient to be published.

Author Response

Dear Reviewer, 

Thank you for taking time to revise the manuscript. 

Best regards,